# The Factors Influencing the Incidence, Persistence, and Severity of Symptoms After SARS-CoV-2 Infection in Chinese Adults: A Case–Control Study

**DOI:** 10.3390/tropicalmed10070185

**Published:** 2025-06-30

**Authors:** Weixiao Wang, Runjie Qi, Siyue Jia, Zhihang Peng, Hongxing Pan, Ming Xu, Yuanbao Liu, Xiaoqiang Liu, Qing Wang, Li Zhang, Jihai Tang, Hao Yang, Pengfei Jin, Simin Li, Jingxin Li

**Affiliations:** 1National Vaccine Innovation Platform and Department of Epidemiology, School of Public Health, Nanjing Medical University, Nanjing 210009, China; 15666069532@163.com (W.W.); qrj1232022@163.com (R.Q.); 2NHC Key Laboratory of Enteric Pathogenic Microbiology, Jiangsu Provincial Center for Disease Control and Prevention, 172 Jiangsu Rd., Nanjing 210009, China; jiasiyue0616@163.com (S.J.); panhongxing@126.com (H.P.); sosolou@jscdc.cn (M.X.); lybaomc@163.com (Y.L.); jpf19891103@163.com (P.J.); 3China Center for Disease Control and Prevention, Beijing 102206, China; zhihangpeng@njmu.edu.cn; 4Yunnan Provincial Center for Disease Control and Prevention, Kunming 650022, China; liuxqms@163.com; 5Chongqing Provincial Center for Disease Control and Prevention, Chongqing 402100, China; cqcdc_mygh@126.com; 6Shandong Provincial Center for Disease Control and Prevention, Jinan 250014, China; zl9127@163.com; 7Anhui Provincial Center for Disease Control and Prevention, Hefei 230601, China; tjh@ahcdc.com.cn; 8Hunan Provincial Center for Disease Control and Prevention, Changsha 410153, China; nicjoe@126.com; 9School of Public Health, Southeast University, Nanjing 210009, China; 13342405350@163.com

**Keywords:** COVID-19, persistent symptoms, severity symptoms, influence factors

## Abstract

Following the emergence of COVID-19, breakthrough SARS-CoV-2 infections have demonstrated substantial heterogeneity in both occurrence and clinical severity. This case–control study aimed to elucidate the factors associated with the incidence, duration, and severity of SARS-CoV-2 symptoms among Chinese adults during the Omicron wave. The analysis was based on data from a national COVID-19 surveillance program encompassing six provinces—Jiangsu, Chongqing, Shandong, Hunan, Anhui, and Yunnan—and included both laboratory-confirmed and clinically diagnosed cases. Data were systematically collected between February and April 2023. For each confirmed case, a matched control was selected through simple random sampling, matched on sex, age (±5 years), and province of residence. Multivariate logistic regression analyses were employed to assess a range of potential determinants, including demographic characteristics, lifestyle behaviors, and pre-existing medical conditions, in relation to the risk of infection, as well as the persistence and severity of symptoms following SARS-CoV-2 breakthrough infection. A total of 10,426 cases and 10,426 matched controls were included in the final analysis. Among the infected individuals, 963 (9.24%) reported persistent symptoms, while 773 (7.41%) experienced moderate-to-severe clinical manifestations. Occasional alcohol consumption, presence of comorbidities, tea and coffee intake, overweight status, and a longer interval since the last vaccination dose were all significantly associated with increased odds of infection (OR > 1, FDR < 0.05). Conversely, weekly alcohol consumption and smoking were associated with a decreased risk (OR < 1, FDR < 0.05). Female sex was significantly associated with both persistent and moderate-to-severe symptoms. Additional risk factors for prolonged or severe symptoms included older age, being underweight or overweight, a history of immunotherapy, coffee consumption, and the presence of comorbidities. These findings underscore the multifactorial nature of SARS-CoV-2 infection outcomes and highlight the interplay between host characteristics and behavioral factors. The results support the development of personalized prevention strategies aimed at reducing the clinical burden and long-term impact of COVID-19.

## 1. Introduction

The COVID-19 pandemic, as a major public health crisis, has had profound adverse impacts on human health, the global economy, and societal functioning. During 2020–2021, approximately 16 million deaths were attributed to COVID-19, and global life expectancy declined by 1.6 years [1]. While most individuals recover within days of infection, a subset experience symptoms that persist for weeks or longer—commonly referred to as “post-acute COVID-19 symptoms” or “long COVID” [2]. The risk and severity of symptoms vary among populations with different immunization statuses. Complete COVID-19 vaccination has been shown to reduce the risk of both infection and the development of long COVID symptoms [3,4]. Several studies have identified risk factors for long COVID. For example, large cohort studies conducted in China and the United Kingdom have demonstrated that female sex, older age, and pre-existing comorbidities are associated with an increased risk of long COVID [5,6]. A systematic review and meta-analysis of 41 studies further found that female sex, advanced age, higher body mass index (BMI), and smoking were significantly associated with symptoms persisting for three months or longer following the acute phase of COVID-19 infection [7].

In China, the COVID-19 vaccine coverage rate has exceeded 90% [8]. However, due to ongoing viral evolution, the threat of SARS-CoV-2 has not been fully mitigated. Notably, following the lifting of strict public health measures in late 2022, a widespread outbreak dominated by the Omicron subvariants BA.5.2 and BF.7 occurred, leading to a large wave of infections across the country. To investigate the determinants of breakthrough infection, symptom persistence, and moderate-to-severe clinical manifestations, we conducted a case–control study. We collected data on disease onset, clinical symptoms, lifestyle behaviors, and demographic characteristics from affected individuals during this period.

## 2. Method

### 2.1. Data Sources

This retrospective analysis utilized data from an established COVID-19 epidemiological survey conducted between 4 February and 10 April 2023. The study protocol was approved by the Ethics Review Committee of the Chinese Center for Disease Control and Prevention. The participants included adult permanent residents (≥18 years) recruited from six provinces in China: Jiangsu, Chongqing, Shandong, Hunan, Anhui, and Yunnan. COVID-19 vaccination records were obtained from the National Integrated Vaccination Service Management Information System. Eligible participants were enrolled using a previously described protocol. Briefly, individuals from heterologous vaccine trial cohorts (ClinicalTrials.gov identifiers: NCT04892459, NCT04952727, NCT05043259, NCT05303584, NCT05204589) were matched 1:4 with community-based controls [9]. All participants were invited to participate in a telephone survey. Prior to the commencement of the survey, oral informed consent was obtained from each individual. To meet the research investigation requirements of this study, we designed a standardized questionnaire (Appendix B Method 1). The participants completed a standardized questionnaire through a structured question-and-answer format.

The standardized questionnaire focused on four aspects: (1) Basic demographic information, including residence, height, weight, age, and sex. (2) Lifestyle factors, such as alcohol consumption, smoking, tea consumption, and coffee consumption. Drinking habits were defined as the consumption of beer, liquor, or other alcoholic beverages. Smoking habits were defined as tobacco use, including cigarettes and traditional pipes. A lifestyle habit was considered present if the individual engaged in it at least once every six months. (3) Previous medical history, including comorbidities, allergies, and immunotherapy. The chronic diseases recorded included hypertension, hyperlipidemia, diabetes, stroke, coronary heart disease, and others. (4) COVID-19-related information, including infection status and date, diagnostic method, symptoms, symptom duration, and severity (Appendix B Method 1).

### 2.2. Study Participants

COVID-19 cases were defined as either (1) etiology-confirmed cases, based on positive nucleic acid or rapid antigen test results, or (2) clinical cases presenting at least two typical symptoms (e.g., fever, cough, weakness, fatigue, headache, myalgia, sore throat, rhinitis, dyspnea, nausea, diarrhea, or anorexia) in conjunction with a known COVID-19 exposure. When the case–control cohort was established, one-to-one individual matching was performed, with each case matched to a control participant based on sex, province of residence, and age (±5 years). A sequential matching procedure was employed as follows: for each control, all eligible cases meeting the matching criteria were identified, and one was randomly selected using simple random sampling to minimize potential selection bias. Once matched, the selected case was removed from the case pool to avoid duplicate assignment. The entire matching process was implemented using R software (version 4.4.0), with the sample function employed for random selection. A fixed random seed was set to ensure reproducibility of the results. The cohort selection flow is presented in Figure 1.

### 2.3. Study Design

A case–control design was used in this study. The COVID-19 cases in our study included etiology-confirmed cases based on positive nucleic acid or rapid antigen test results and clinical cases presenting with at least two typical symptoms in combination with a COVID-19 exposure. We further analyzed the factors influencing incidence, persistence, and severity symptoms of SARS-CoV-2 breakthrough infection, including demographic characteristics, lifestyle habits, and medical history.

For the purposes of this study, the frequency of these habits was categorized into three groups: “never”, “occasionally”, and “weekly”. The “never” category was defined as abstaining from the habit for six months or more. The responses “Occasionally”, “Only during specific times”, and “1–3 times a month” were grouped into the “occasionally” category, defined as a frequency greater than once every six months but less than three times per month. The “weekly” category was defined as engaging in the habit at least once per week.

The severity and duration of symptoms were systematically documented. A prior study comprehensively categorized long-term COVID-19 symptoms into nine non-neurological organ systems and nine nervous system-related manifestations [10]. Building upon this framework, the symptoms in our study were classified into five categories: respiratory, gastrointestinal, constitutional, neurological/sensory, and musculoskeletal symptoms. Symptom severity was assessed using the Common Terminology Criteria for Adverse Events (CTCAE) standards (Appendix A Table A1), with moderate-to-severe symptoms defined as grade 2 or higher. For symptom persistence, we adopted the definition from a primary care study on long-term COVID-19 sequelae [2], where post-acute COVID-19 was characterized by symptom persistence exceeding 21 days following disease onset. Accordingly, persistent symptoms in this study were defined as those lasting ≥ 21 days.

### 2.4. Statistical Analyses

To enhance data quality, we excluded samples with missing key variables (including sex, age, height, weight, COVID-19 infection status, post-infection symptoms, lifestyle habits, and self-reported illness status) and identified duplicate entries.

For descriptive statistics of baseline features, continuous variables, such as age and BMI, were summarized as the mean (standard deviation) or the median (interquartile range, IQR), and categorical variables were summarized as frequency (percentage). Independent-samples *t*-tests, chi-square tests, or Fisher’s exact tests were used to assess differences between groups.

Odds ratios (ORs) with corresponding 95% confidence intervals (CIs) were calculated using multivariate logistic regression analyses to identify factors associated with COVID-19 incidence. Subsequently, multivariate logistic regression models were employed to examine the relationships between COVID-19 outcomes (including moderate-to-severe symptoms and persistent symptoms) and various factors, such as demographic characteristics, lifestyle factors, and comorbidities. Statistical significance was determined at *p* < 0.05, with *p*-values adjusted using the False Discovery Rate (FDR) method. The analysis included patients with varying clinical presentations of COVID-19, and ORs with CIs were used to quantify the strength of associations between predictors and COVID-19-related outcomes.

A Venn diagram was used to illustrate the overlap among the five symptom categories in relation to moderate-to-severe and persistent symptoms. Two-sided *p*-values were reported for all statistical tests, and a *p*-value less than 0.05 was considered statistically significant. All statistical analyses were performed using R software (version 4.4.0).

## 3. Results

### 3.1. Description of Study Population

We included a total of 10,462 patients infected with COVID-19 and 10,462 uninfected controls. As shown in the baseline characteristics (Table 1), the two groups were balanced in terms of sex and age. The mean BMI was 23.88 ± 3.39 in the case group and 23.72 ± 3.52 in the control group. Among the participants, 3278 (31.44%) in the case group and 4002 (38.38%) in the control group were smokers. Regarding alcohol consumption, 3740 participants (35.75%) in the case group and 3904 (37.32%) in the control group reported alcohol use. In terms of tea consumption, 4538 participants (43.38%) in the case group and 4099 (39.18%) in the control group reported habitual tea drinking. Additionally, 700 (6.69%) in the case group and 552 (5.28%) in the control group reported habitual coffee consumption. Regarding comorbidities, 3281 participants (31.47%) in the case group and 2613 (25.06%) in the control group had at least one comorbidity. In the case group, 2333 (22.38%) had a single comorbidity, 697 (6.69%) had two, and 251 (2.41%) had three or more. In the control group, 1985 (19.04%) had a single comorbidity, 484 (4.64%) had two, and 144 (1.38%) had three or more. The five most common comorbidities across both groups were hypertension (2807 [13.46%]), stroke (498 [2.39%]), hyperlipidemia (173 [0.83%]), myocardial infarction (131 [0.63%]), and diabetes (125 [0.60%]).

### 3.2. COVID-19 Symptoms

According to the surveillance data from the Chinese Center for Disease Control and Prevention, BA.5.2 and BF.7 were the predominant variants circulating in China at the end of 2022 [11]. In this study, we investigated 19 symptoms associated with COVID-19 during this period (Table 2), including constitutional symptoms (9284 [88.74%]), respiratory symptoms (7470 [71.40%]), musculoskeletal symptoms (4674 [44.68%]), neurological and sensory symptoms (4218 [40.32%]), and gastrointestinal symptoms (2148 [20.53%]). Among individual symptoms, the highest correlation was found between hypogeusia and hyposmia (Pearson correlation = 0.57, *p* < 0.0001). After clustering, the Pearson correlation between symptoms was less than 0.24 (Appendix C Figure A1). There were 963 (9.21%) participants who experienced persistent symptoms. The most common persistent symptom categories were respiratory symptoms, affecting 600 individuals; followed by constitutional symptoms, affecting 282 individuals; neurological or sensory symptoms, affecting 176 individuals; musculoskeletal symptoms, affecting 82 individuals; and gastrointestinal symptoms, affecting 64 individuals. There were 773 (7.39%) individuals with moderate-to-severe symptoms. The most common symptoms were respiratory symptoms in 416 individuals, musculoskeletal symptoms in 218 participants, followed by constitutional symptoms in 198 participants, neurological or sensory symptoms in 155 participants, and gastrointestinal symptoms in 52 participants (Figure 2 and Table 2).

### 3.3. Factors Associated with the Incidence of COVID-19

In the multivariable regression analysis of factors associated with COVID-19 incidence (Table 3), smoking (OR: 0.69, FDR < 0.001) and weekly alcohol consumption (OR: 0.90, FDR = 0.030) were associated with a lower risk compared to never smoking or alcohol consumption. Conversely, factors associated with an increased COVID-19 incidence included a higher BMI (OR: 1.08, FDR = 0.010), occasional alcohol consumption (OR: 1.15, FDR < 0.001), occasional coffee consumption (OR: 1.19, FDR = 0.010), weekly coffee consumption (OR: 1.41, FDR = 0.040), occasional tea consumption (OR: 1.34, FDR < 0.001), weekly tea consumption (OR: 1.29, FDR < 0.001), and history of allergies (OR: 1.37, FDR < 0.001) compared to their respective reference groups. Additionally, individuals with chronic conditions, such as hypertension only (OR: 1.27, FDR < 0.001), hyperlipemia only (OR: 1.70, FDR = 0.009), having one chronic disease (OR: 1.29, FDR < 0.001), having two chronic diseases (OR: 1.57, FDR < 0.001), or having three or more chronic diseases (OR: 1.85, FDR < 0.001), had a higher likelihood of infection compared to those without chronic diseases.

### 3.4. Factors Associated with Symptom Categories

Multivariate analysis was conducted to examine factors influencing five categories of symptoms, focusing on persistent symptoms (lasting ≥ 21 days) or moderate-to-severe symptoms (severity of grade 2 or higher) (Figure 3).

In the analysis of respiratory symptoms, smoking and male sex were significantly associated with a lower likelihood of persistent symptoms. Smokers had a lower risk compared to non-smokers (OR: 0.37; FDR < 0.001), and males had a lower risk than females (OR: 0.80, FDR = 0.009). A similar trend emerged for more severe respiratory symptoms (grade 2 or higher). Smokers had a lower risk (OR: 0.58, FDR < 0.001), and males had a lower risk than females (OR: 0.78, FDR = 0.045). Conversely, two or more comorbidities were significantly associated with an increased risk of both symptom categories compared to those without these factors. Additional factors specifically associated with respiratory persistent symptoms included BMI ≥ 28 (OR: 1.31, FDR = 0.009), weekly coffee consumption (OR: 1.85, FDR = 0.002), coronary artery disease (CAD) (OR: 2.09, FDR = 0.020), and the presence of any comorbidity (OR: 1.30, FDR = 0.003). A history of allergies (OR: 1.86, FDR < 0.001) and immunotherapy (OR: 2.26, FDR = 0.020) were also associated with an increased risk. Occasional tea consumption was negatively associated with persistent symptoms (OR: 0.80, FDR = 0.020). The significant risk factors for moderate-to-severe respiratory symptoms included occasional coffee consumption (OR: 1.60, FDR = 0.004) and a history of stroke (OR: 2.76, FDR = 0.020) (Figure 3A).

For constitutional symptoms (Figure 3B), age was identified as a risk factor, with older individuals having a higher likelihood of experiencing persistent symptoms (OR: 1.03, FDR < 0.001). A history of allergies was associated with an increased risk of persistent symptoms (OR: 1.49, FDR = 0.020) compared to those without a history of allergies. In contrast, male sex (OR: 0.66, FDR < 0.001) and smoking (OR: 0.63, FDR = 0.009) were negatively associated with persistent symptoms. The risk factors for persistent symptoms included multiple chronic diseases, CAD (OR: 3.18, FDR < 0.001), hyperlipidemia (OR: 3.05, FDR = 0.002), coffee consumption (occasional vs. never: OR: 1.77, FDR = 0.005; weekly vs. never: OR: 2.28, FDR = 0.010). For moderate-to-severe symptoms, the risk factors included immunosuppressive therapy (OR: 5.28, FDR < 0.001).

For gastrointestinal symptoms, the risk factors for symptoms lasting ≥21 days included older age (OR: 1.04, FDR < 0.001), a history of allergies (OR: 2.46, FDR = 0.004), CAD (OR: 5.17, FDR = 0.001), and weekly coffee consumption (OR: 6.20, FDR < 0.001). The only significant risk factor for moderate-to-severe gastrointestinal symptoms was underweight (BMI < 18.5) (OR: 2.40, FDR = 0.047) (Figure 3C).

In the multivariate analysis of neurological and sensory symptoms, male sex was found to be a protective factor. Compared to females, males had a lower likelihood of experiencing persistent symptoms (OR: 0.61, FDR = 0.002) and a lower likelihood of more severe symptoms (grade 2 or higher) (OR: 0.64, FDR = 0.030). A history of allergies was associated with an increased risk of both persistent symptoms (OR: 1.96, FDR < 0.001) and moderate-to-severe symptoms (OR: 1.68, FDR = 0.049) compared to individuals without a history of allergies. The risk factors for persistent symptoms included a history of allergies, overweight/obesity (BMI ≥ 24), CAD, stroke, weekly coffee consumption, and having one or more comorbidities. Additionally, having three or more comorbidities was associated with an increased risk of moderate-to-severe symptoms (Figure 3D).

For musculoskeletal symptoms, male sex was again a protective factor. Compared to females, males had a lower likelihood of experiencing persistent symptoms (OR: 0.55, FDR = 0.020). The risk factors for both symptom categories included CAD, immunosuppressive therapy, and chronic disease with two comorbidities compared to those without these factors. Chronic disease with one or more comorbidities was a significant risk factor, specifically for moderate-to-severe musculoskeletal symptoms. Obesity (BMI ≥ 28) was positively correlated with persistent symptoms, while overweight status (24 ≤ BMI < 28) was associated with an increased risk of moderate-to-severe symptoms. Weekly tea consumption was also associated with an increased risk of moderate-to-severe musculoskeletal symptoms.

## 4. Discussion

In this study, we conducted a multi-regional, large-scale retrospective case–control analysis to explore the factors associated with COVID-19 infection and symptoms in China. Our findings demonstrated that during the predominance of the BA.5.2 and BF.7 variants at the end of 2022, 9% of the participants with infection experienced persistent symptoms, while 7% experienced moderate-to-severe symptoms. Among the persistent symptoms, the most frequently reported were cough, fatigue, hypogeusia, myalgia, and anorexia. Similarly, the most common moderate-to-severe symptoms included cough, fatigue, myalgia, pharyngalgia, and headache. These findings are consistent with observations reported in previous studies [10,12,13].

In our analysis of factors influencing susceptibility to COVID-19, both smoking and weekly alcohol consumption emerged as protective factors. However, these findings do not imply that smoking or excessive alcohol consumption should be encouraged during the pandemic, as such behaviors are associated with numerous chronic diseases. Further mechanistic studies are required to validate these observations. Conversely, pre-existing chronic conditions, including hypertension and hyperlipidemia, were associated with an elevated risk of COVID-19 infection. This aligns with prior research indicating that comorbidities increase the likelihood of developing long COVID [14,15]. Notably, a dose–response relationship was observed, with higher numbers of comorbidities corresponding to greater infection risk. Additionally, individuals with a history of allergies demonstrated increased susceptibility to COVID-19, potentially attributable to underlying immune dysregulation [16,17]. Tea consumption was associated with an increased risk of breakthrough infection. While some studies suggest green tea may confer protection against the Omicron variant [18], others report no correlation [19]. The observed risk in our study may be partially explained by the social interactions typically accompanying tea consumption in China, which could enhance interpersonal contact and transmission opportunities. Similarly, coffee consumption was identified as a risk factor for SARS-CoV-2 infection, consistent with findings from a Japanese cohort study where higher coffee intake correlated with increased Omicron infection risk among triple-vaccinated healthcare workers [19].

In analyzing symptom-specific associations, most trends mirrored those observed for infection risk. However, certain factors exhibited divergent effects across symptom categories. Male sex consistently served as a protective factor against both persistent and moderate-to-severe symptoms across nearly all categories. This may reflect sex-based differences in immune responses: females exhibit stronger adaptive immunity and heightened proinflammatory cytokine production [20], potentially exacerbating symptom severity and duration during breakthrough infections. Advanced age emerged as a risk factor for persistent constitutional, gastrointestinal, and musculoskeletal symptoms, likely due to age-related immune decline impairing viral clearance and prolonging recovery. Comorbidities were risk factors for most symptom categories except gastrointestinal manifestations, with risk escalating alongside comorbidity burden. These findings are corroborated by a meta-analysis of 41 studies [6]. Contrary to most published findings, smoking was not associated with symptom persistence or severity in our analysis and even appeared protective against respiratory symptoms. While preliminary data suggest nicotine receptor interactions might underlie this observation [21], such hypotheses remain unconfirmed and do not justify smoking initiation or continuation. Coffee consumption, particularly weekly intake, was associated with the most persistent symptoms. Mechanistically, daily consumption exceeding 200 mL correlates with elevated proinflammatory markers (e.g., CRP, IL-6, TNF-α) [22,23], which may impair antiviral immunity. The last inactivated vaccine dose exceeding 9 months was identified as a risk factor for persistent symptoms but served as a protective factor against symptom severity. This is because neutralizing antibody titers decline to pre-vaccination levels 8 months post-booster [24]. Studies on immunity persistence after original COVID-19 vaccination indicate that memory B and T cells decrease six months post-vaccination, particularly against the Omicron variant [25,26]. Consequently, during breakthrough infections, the immune response is weaker, resulting in reduced inflammatory factor secretion and milder symptoms compared to those vaccinated within 9 months, though symptoms may persist longer. Although BMI was not consistently associated with infection or general symptom risk, both low (<18.5) and high BMI (≥28) were linked to higher risks of persistent and severe symptoms in breakthrough cases, relative to individuals with BMI values of 18.5–28.

Existing research indicates that certain risk factors demonstrate consistent effects across multiple COVID-19 variants. Studies tracking the viral evolution from Alpha to Beta to Gamma variants in 2020 have identified several persistent risk factors for long COVID, including advanced age [27,28], female sex [29,30], and pre-existing comorbidities [31,32,33], such as hypertension, hyperlipidemia, and diabetes. These findings suggest that these demographic and clinical characteristics maintain their predictive value regardless of the specific viral variant, highlighting consistent patterns in COVID-19 susceptibility throughout the pandemic’s progression.

Our study has several limitations. First, the retrospective design may introduce recall bias, as participants might inaccurately recall details regarding their lifestyle, medical history, and symptoms. Second, symptom data were self-reported, which could lead to variability in reporting accuracy and consistency across individuals. Third, because of the limited availability of rapid antigen tests or nucleic acid tests during outbreaks, only a small percentage of reported COVID-19 cases were confirmed by etiology. Clinical cases of COVID-19 confirmed by symptoms and exposure history may be misdiagnosed, and our study may have missed some asymptomatic cases. Fourth, some influencing factors were not included in the questionnaire, such as dietary pattern, occupation, ethnicity, and education level. Despite these limitations, our study provides valuable insights into the common factors influencing COVID-19 infection and its associated symptoms among Chinese adults.

In conclusion, our findings indicate that coffee consumption is associated with an elevated risk of infection and more persistent symptoms. Additionally, comorbidities and female sex were identified as significant risk factors. While abnormal BMI was not consistently associated with COVID-19 infection or symptoms, it was not found to be a protective factor. Although the COVID-19 epidemic has subsided, SARS-CoV-2 continues to evolve and mutate. To better prepare for future health emergencies, further research is needed to investigate the factors influencing SARS-CoV-2 outcomes in individuals with diverse immune statuses.

## Figures and Tables

**Figure 1 tropicalmed-10-00185-f001:**
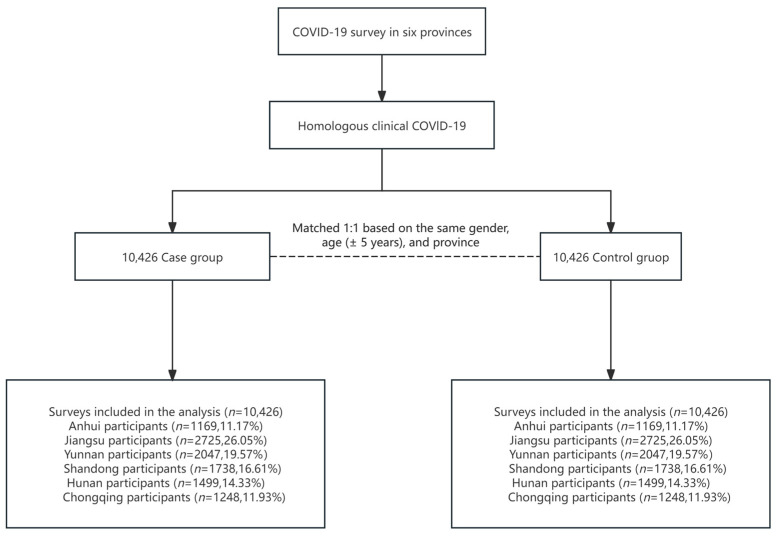
Selection of cohorts with three doses of an inactivated vaccine from different regions of China prior to the pandemic.

**Figure 2 tropicalmed-10-00185-f002:**
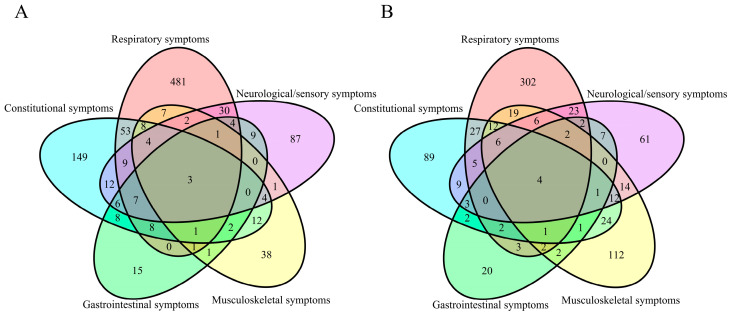
Coexistence of symptom categories: (**A**) symptoms persisting for ≥21 days; (**B**) symptoms with severity ≥ grade 2.

**Figure 3 tropicalmed-10-00185-f003:**
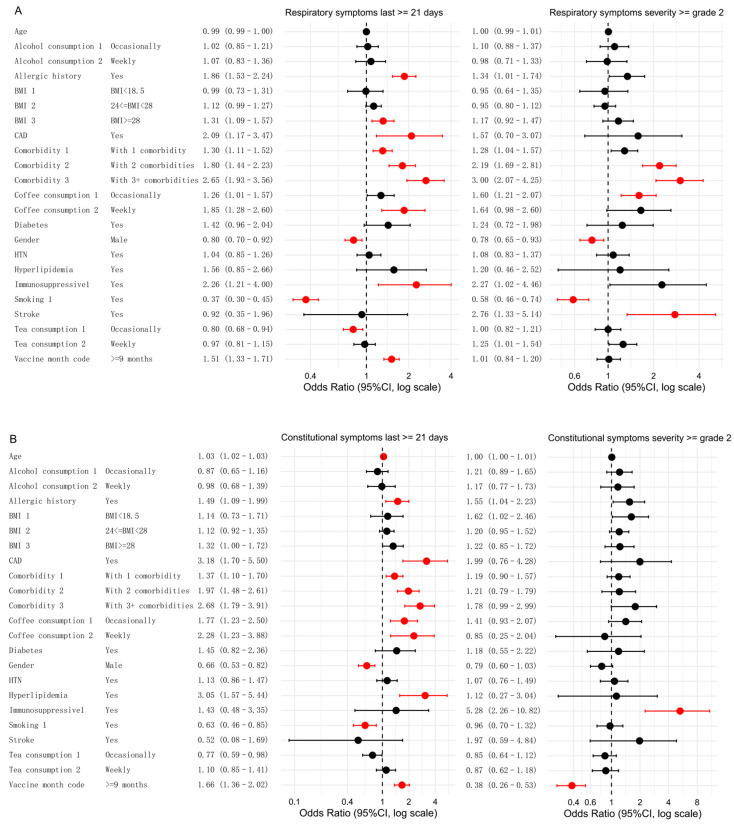
Associations between different influencing factors and both persistent and moderate-to-severe symptoms. Multivariate analyses were conducted to assess the relationships for five symptom categories: respiratory symptoms (**A**), constitutional symptoms (**B**), gastrointestinal symptoms (**C**), neurological or sensory symptoms (**D**), and musculoskeletal symptoms (**E**). Significant factors (FDR < 0.05) are highlighted in red, while non-significant factors are shown in black. Note: For gastrointestinal symptoms of grade ≥2, the number of participants with coronary artery disease (CAD) and hyperlipidemia was 142 and 143, respectively. Because of insufficient sample size, the statistical results (*p* = 0.98 for both) were not meaningful and would compromise the clarity of the forest plot; therefore, these variables are indicated as “—”.

**Table 1 tropicalmed-10-00185-t001:** Baseline characteristics of the participants.

	Control Group (*n* = 10,462, %)	Case Group (*n* = 10,462, %)	*p*
Age group (year)			
18–29	608 (5.83)	608 (5.83)	0.990
30–39	1285 (12.32)	1285 (12.32)	
40–49	1505 (14.44)	1548 (14.85)	
50–59	2480 (23.79)	2469 (23.68)	
60–69	2601 (24.95)	2563 (24.58)	
≥70	1947 (18.68)	1953 (18.73)	
Mean age, years	55.28 ± 14.92	55.20 ± 14.90	0.690
Sex			
Male	6686 (64.13)	6686 (64.13)	1.000
Mean BMI, kg/m^2^	23.72 ± 3.52	23.88 ± 3.39	<0.001
BMI, kg/m^2^			
≥18.5 and <24	5282 (52.03)	5024 (49.16)	<0.001
<18.5	444 (4.37)	383 (3.75)	
≥24 and <28	3468 (34.16)	3742 (36.61)	
≥28	958 (9.44)	1071 (10.48)	
Interval from last dose vaccination to 1 November 2022 (days)	252.00 (247.00, 256.00)	252.00 (247.00, 257.00)	0.760
Interval from last dose vaccination to 1 November 2022 (months)			
5–8 months	8831 (84.70)	8509 (81.61)	<0.001
≥9 months	1595 (15.30)	1917 (18.39)	
Smoking			
No	6424 (61.62)	7148 (68.56)	<0.001
Yes	4002 (38.38)	3278 (31.44)	
Alcohol consumption			
Never/hardly ever	6522 (62.56)	6686 (64.13)	<0.001
Weekly	2198 (21.08)	2365 (22.68)	
Occasionally	1706 (16.36)	1375 (13.19)	
Tea consumption			
Never/hardly ever	6327 (60.68)	5888 (56.47)	<0.001
Weekly	2072 (19.87)	2374 (22.77)	
Occasionally	2027 (19.44)	2164 (20.76)	
Coffee consumption			
Never/hardly ever	9874 (94.71)	9726 (93.29)	<0.001
Weekly	480 (4.60)	594 (5.70)	
Occasionally	72 (0.69)	106 (1.02)	
Allergic history	377 (3.62)	554 (5.31)	<0.001
Immunological therapy history	24 (0.23)	44 (0.42)	0.020
Types of comorbidities			
No	7813 (74.94)	7145 (68.53)	<0.001
Only hypertension	1300 (12.47)	1507 (14.45)	
Only diabetes	48 (0.46)	77 (0.74)	
Only stroke	239 (2.29)	259 (2.48)	
Only hyperlipemia	78 (0.75)	95 (0.91)	
Only coronary heart disease	63 (0.60)	68 (0.65)	
Number of comorbidities			
No	7813 (74.94)	7145 (68.53)	<0.001
One comorbidity	1985 (19.04)	2333 (22.38)	
Two comorbidities	484 (4.64)	697 (6.69)	
Three or more comorbidities	144 (1.38)	251 (2.41)	

Data are *n* (%), mean (SD), or median (IQR), BMI: body mass index.

**Table 2 tropicalmed-10-00185-t002:** COVID-19 symptoms.

Symptoms	Total (*n* = 10,462)	Symptoms Lasting ≥ 21 Days (*n* = 963, 9%)	Symptoms Severity ≥ Grade 2 (*n* = 773, 7%)
Respiratory symptoms	7470 (71.40)	600 (62.31)	416 (53.82)
cough	6100 (58.30)	566 (58.77)	259 (33.51)
pharyngalgia	3311 (31.65)	41 (4.25)	142 (18.37)
rhinitis	1953 (18.67)	36 (3.73)	56 (7.25)
dyspnea	332 (3.17)	49 (5.09)	42 (5.43)
Constitutional symptoms	9284 (88.74)	282 (29.28)	198 (25.61)
fatigue	6662 (63.68)	279 (28.97)	197 (25.49)
chill	1384 (13.23)	14 (1.45)	2 (0.26)
fever	7114 (68.00)	1 (0.10)	-
Neurological and sensory symptoms	4218 (40.32)	176 (18.28)	155 (20.05)
headache	3092 (29.55)	36 (3.74)	94 (12.16)
hypogeusia	1618 (15.47)	92 (9.55)	51 (6.60)
hyposmia	802 (7.67)	59 (6.12)	25 (3.23)
eye discomfort	151 (1.44)	10 (1.04)	5 (0.65)
attention disorders	143 (1.37)	23 (2.39)	4 (0.52)
alopecia	50 (0.48)	16 (1.66)	0 (0.00)
Musculoskeletal symptoms	4674 (44.68)	82 (8.52)	218 (28.21)
myalgia	4374 (41.81)	68 (7.06)	186 (24.06)
ostalgia	886 (8.47)	26 (2.70)	74 (9.57)
Gastrointestinal symptoms	2148 (20.53)	64 (6.65)	52 (6.73)
anorexia	1853 (17.71)	60 (6.23)	35 (4.53)
diarrhea	255 (2.43)	4 (0.42)	10 (1.29)
emesis	219 (2.09)	1 (0.10)	9 (1.16)
nausea	270 (2.58)	6 (0.62)	2 (0.26)

**Table 3 tropicalmed-10-00185-t003:** Multivariate analysis of the incidence of COVID-19.

Characteristic	OR (95% CI)	*p*	FDR
BMI, kg/m^2^			
18.5 ≤ BMI < 24	1.00 (Reference)		
BMI < 18.5	0.92 (0.80~1.06)	0.269	0.283
24 ≤ BMI < 28	1.08 (1.02~1.15)	0.008	0.013
BMI ≥ 28	1.07 (0.97~1.18)	0.150	0.164
Smoking			
No	1.00 (Reference)		
Yes	0.69 (0.64~0.74)	<0.001	<0.001
Alcohol consumption			
Never/hardly ever	1.00 (Reference)		
Occasionally	1.15 (1.07~1.24)	<0.001	<0.001
≥1/week	0.90 (0.82~0.98)	0.021	0.032
Tea consumption			
Never/hardly ever	1.00 (Reference)		
Occasionally	1.33 (1.24~1.43)	<0.001	<0.001
≥1/week	1.29 (1.20~1.39)	<0.001	<0.001
Coffee consumption			
Never/hardly ever	1.00 (Reference)		
Occasionally	1.19 (1.05~1.35)	0.006	0.011
≥1/week	1.41 (1.04~1.93)	0.024	0.035
Types of comorbidities			
None	1.00 (Reference)		
Hypertension	1.27 (1.17~1.38)	<0.001	<0.001
Hyperlipemia	1.70 (1.18~2.46)	0.004	0.009
Diabetes	1.17 (0.98~1.40)	0.080	0.100
Coronary heart disease	1.32 (0.97~1.79)	0.070	0.093
Stroke	1.19 (0.84~1.69)	0.310	0.310
Number of comorbidities			
None	1.00 (Reference)		
One comorbidity	1.29 (1.20~1.38)	<0.001	<0.001
Two comorbidities	1.57 (1.39~1.77)	<0.001	<0.001
Three or more comorbidities	1.85 (1.50~2.28)	<0.001	<0.001
Allergic history			
No	1.00 (Reference)		
Yes	1.37 (1.20~1.57)	<0.001	<0.001
Immunosuppressive			
No	1.00 (Reference)		
Yes	1.48 (0.89~2.49)	0.130	0.153
Interval from last dose vaccination to 1 November 2022 (months)			
5–8 months	1.00 (Reference)		
≥9 months	1.28 (1.19~1.38)	<0.001	<0.001

BMI: body mass index; OR: odds ratio; CI: confidence interval; FDR: *p*-values were adjusted using the FDR method.

## Data Availability

According to the national policies of the People’s Republic of China, the data of this research will not be made public.

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
