# Peer review of "The Factors Influencing the Incidence, Persistence, and Severity of Symptoms After SARS-CoV-2 Infection in Chinese Adults: A Case–Control Study"

_tropicalmed, 2025, doi:10.3390/tropicalmed10070185_

Round 1
Reviewer 1 Report
Comments and Suggestions for Authors
This case-control study aimed to explore the factors 23 influencing the incidence, persistence, and severity of symptoms in SARS-CoV-2 infections among Chinese adults usingdata from several vaccine trial cohorts and community based control data in six provinces in China. Data was collected via survey, administered on a group of individuals with confirmed COVID 19 infections and those without and had many data available for analysis. Methodology for long-COVID symptoms was described.
Odds rations with 95% CI were calculated using multivariate regression analysis. Statistical significance through false discovery rate method. Venn diagrem used and two-sided p-values were reported for tests.
10,462 COVID-19 ifected patients and un-infected controls were reviewed.
Data showed multiple health indicators to be assocated with more covid infection.
Surprising findings include tobacco, and occasional alcohol use associated with less infection.
symptoms lasting >21 days or being >2 grade were described.
In the multivariate analsysis smoking was protective of COVID incidence. Smoking and male sex were significantly associated with a lower likelihood of persistent symptoms.
Conclusion was that during the BA.5.2 and BF.7 variant part of the epidemic at the end of 2022, 9% of participants experienced chronic symptoms and 7% developed moderate to severe symptoms. Smoking and weekly alcohol consumption were associated with protective effect. Male sex appeared protective as well for persistent and moderate to severe infection.
This is an interesting and well thought out analysis. There are some grammatical improvements and stylistic changes that an editor could help with. Otherwise this is a great analysis.
Comments on the Quality of English LanguageThere are quite a few sentences that are incomplete and have stylistic errors. Example line 37 is an incomplete sentence. Line 48 is a repeated word.
Author Response
Comments 1:[This is an interesting and well thought out analysis. There are some grammatical improvements and stylistic changes that an editor could help with. Otherwise this is a great analysis.
Comments on the Quality of English Language:There are quite a few sentences that are incomplete and have stylistic errors. Example line 37 is an incomplete sentence. Line 48 is a repeated word.]
Response 1:[We greatly appreciate your positive feedback on our analysis. We have meticulously proofread the entire manuscript with particular attention to grammar and style, and have implemented comprehensive revisions to enhance clarity and readability throughout.]
Reviewer 2 Report
Comments and Suggestions for Authors
Thank you for the opportunity to review this manuscript.The study is well-documented and offers significant insights regarding the severity, incidence of COVID-19 symptoms and risk factors among adults infected during the Omicron wave.
Comments and Suggestions:
In the abstract, consider removing the subheadings such as “Background,” “Methods,” and “Findings” for a more concise and fluid structure.
In the first paragraph of the abstract, it would be helpful to specify that the study focuses on infections that occurred during the Omicron wave.
It would be useful to elaborate on what is meant by a “standardized questionnaire” – specifically, to clarify that it was designed specifically for this study.
Considering the focus on lifestyle factors, why were more variables not included? For example, dietary patterns could provide relevant insights.
E-cigarette use is mentioned in the early part of the manuscript but is not addressed later in the analysis – this aspect should be further discussed
Would it be useful to include a comparison of the findings with data from previous COVID-19 waves? This could help contextualize the results and strengthen the conclusions.
Lines 321 and 333 require correction of writing.
Author Response
Response 1:[We appreciate reviewers’ comments. We have revised the abstract by removing all subheadings and carefully integrating the information into a continuous narrative.]
Comments 2:[In the first paragraph of the abstract, it would be helpful to specify that the study focuses on infections that occurred during the Omicron wave.]
Response 2:[We appreciate the reviewer's valuable comments. We have revised the final sentence of the abstract's background section to address this point. The updated text (Lines 22-25) now reads as follows: Following the emergence of COVID-19, breakthrough SARS-CoV-2 infections exhibited marked heterogeneity in occurrence and severity. This case-control study examined factors linked to the incidence, duration, and clinical outcomes of infections among Chinese adults during the Omicron wave.]
Comments 3:[It would be useful to elaborate on what is meant by a “standardized questionnaire” – specifically, to clarify that it was designed specifically for this study.]
Response 3:[We appreciate the reviewer's comment on the questionnaire description. The text has been revised (Lines 82-84) to explicitly state that this was a study-specific questionnaire. ]
Comments 4:[Considering the focus on lifestyle factors, why were more variables not included? For example, dietary patterns could provide relevant insights.]
Response 4:[We sincerely appreciate the reviewer's insightful observation concerning the consideration of dietary patterns and additional lifestyle variables in our analysis. Given the constraints of the standardized surveillance questionnaire deployed across six provinces, we prioritized factors that could be reliably captured with concise, non-burdensome questions suitable for rapid data collection during a public health emergency. As this was a retrospective study, participants would have had difficulty providing accurate data on their dietary patterns. We acknowledge that the absence of detailed dietary data is a limitation and agree that future, prospectively designed studies incorporating validated dietary assessment tools would be valuable to explore this relationship more thoroughly. Therefore, we have added content related to this aspect in the limitations section of the discussion. After modification, it is as follows: some influencing factors are not included in the questionnaire, such as dietary pattern, occupation, ethnicity, and education level. (Lines 366-368)]
Comments 5:[E-cigarette use is mentioned in the early part of the manuscript but is not addressed later in the analysis – this aspect should be further discussed.]
Response 5:[We appreciate the reviewer's astute observation concerning the reference to e-cigarette use without accompanying analysis. Following data verification, we confirmed that our study exclusively captured tobacco use. We have therefore amended the definition of smoking in the manuscript to accurately represent this aspect of our methodology. (Line 91)]
Comments 6:[Would it be useful to include a comparison of the findings with data from previous COVID-19 waves? This could help contextualize the results and strengthen the conclusions. ]
Response 6:[We appreciate the reviewer's valuable suggestions. We fully agree with this perspective and have now compared the main findings of our study with data from previous COVID-19 epidemic waves. These comparisons have been incorporated into the Discussion section (Lines 350-357) as follows:
Existing research indicates that certain risk factors demonstrate consistent effects across multiple COVID-19 variants. Studies tracking the viral evolution from Alpha to Beta to Gamma variants in 2020 have identified several persistent risk factors for long COVID, including advanced age, female sex, and pre-existing comorbidities such as hypertension, hyperlipidemia, and diabetes. These findings suggest that these demographic and clinical characteristics maintain their predictive value regardless of the specific viral variant, highlighting consistent patterns in COVID-19 susceptibility throughout the pandemic's progression.]
Comments 7:[Lines 321 and 333 require correction of writing.]
Response 7:[We appreciate the reviewer's careful observation. The error has been corrected in the manuscript.]